# Phylogenetic Analysis of *Chandipura virus*: Insights from a Preliminary Genomic Study

**DOI:** 10.3390/ijms26031021

**Published:** 2025-01-25

**Authors:** Marta Giovanetti, Valeria Micheli, Alessandro Mancon, Davide Mileto, Alberto Rizzo

**Affiliations:** 1Department of Sciences and Technologies for Sustainable Development and One Health, Campus Bio-Medico University, 00128 Rome, Italy; giovanetti.marta@gmail.com; 2Oswaldo Cruz Institute, Oswaldo Cruz Foundation, Belo Horizonte 30190-002, MG, Brazil; 3Laboratory of Clinical Microbiology, Virology and Bioemergencies, Luigi Sacco Hospital, ASST Fatebenefratelli Sacco, 20157 Milan, Italy; micheli.valeria@asst-fbf-sacco.it (V.M.); mancon.alessandro@asst-fbf-sacco.it (A.M.); mileto.davide@asst-fbf-sacco.it (D.M.)

**Keywords:** *Chandipura virus*, emerging pathogen, phylogenesis, surveillance, one health

## Abstract

*Chandipura virus* (CHPV) is an arthropod-borne virus linked to encephalitis in humans, primarily in India. Its evolutionary dynamics and transmission pathways remain poorly understood due to limited genomic data. This study analyzed 23 publicly available CHPV genomes, including isolates from humans, sandflies, and a hedgehog, retrieved from GenBank. Phylogenetic analyses were conducted to explore host-specific and geographic evolutionary patterns. Phylogenetic analysis revealed distinct evolutionary lineages. Human-derived genomes collected in India between 2003 and 2024 formed a well-supported monophyletic clade, suggesting a unique evolutionary lineage. In contrast, sandfly-derived genomes exhibited diverse clustering patterns. Notably, Kenyan sandfly isolates from 2016–2017 were phylogenetically closer to human-derived sequences, suggesting possible shared evolutionary pressures. These findings provide preliminary insights into CHPV evolution and emphasize the need for enhanced genomic surveillance in both human and non-human populations. Expanding genomic data is essential to validate these observations and inform public health strategies.

## 1. Introduction

The *Chandipura virus* (CHPV), a member of the genus *Vesiculovirus* in the *Rhabdoviridae* family, was first isolated in India in 1965. It is an arthropod-borne, single-stranded RNA virus known for its neurotropic nature, enabling it to cross the blood-brain barrier and invade the central nervous system (CNS) [1,2]. This characteristic makes CHPV capable of causing severe neurological complications, including encephalitis, particularly in children under 15 years of age in India [3]. Since its discovery, recurrent outbreaks have been reported exclusively in India, where the virus remains endemic.

CHPV infection presents with a wide spectrum of symptoms, ranging from influenza-like illness to severe neurological complications. The disease can rapidly progress to acute encephalitis syndrome (AES), characterized by altered mental status, seizures, and coma. In severe cases, AES is associated with high mortality rates and long-term neurological sequelae, such as cognitive deficits and motor dysfunction in survivors [4,5,6,7]. Despite the severity of the disease, no effective vaccines or antiviral therapies are currently available, leaving supportive care as the primary treatment option [8]. The largest recorded human outbreak of CHPV occurred in India between June and August 2024, with 245 cases of acute encephalitis syndrome (AES) diagnosed, including 64 confirmed CHPV infections [9]. This outbreak highlighted the virus capacity for widespread transmission and its significant public health impact. The diagnostic challenges of CHPV are significant due to its clinical overlap with other arboviral and neurological diseases, limited availability of specific diagnostic tools, and the rapid progression of the disease [3].

Sandflies (Phlebotominae) are recognized as the primary vector of CHPV. Additionally, experimental studies have demonstrated transovarial transmission in *Aedes aegypti*, raising concerns about the virus potential expansion beyond traditional sandfly habitats [10]. CHPV has been detected in phlebotomine sandflies and, less frequently, in mammals such as hedgehogs, supporting the hypothesis that sandflies are the principal vector in India and parts of Africa [11,12,13,14,15].

However, the lack of comprehensive genomic and epidemiological data further complicates understanding of the virus geographic spread and evolutionary dynamics. Due to the paucity of reports describing the evolutionary dynamics of CHPV, this study aims to investigate the evolutionary relationships of CHPV genomes collected from humans and other organisms, using phylodynamic analysis to better understand the virus transmission dynamics and host diversity.

## 2. Results

The phylogenetic analysis included five CHPV genomes isolated from Homo sapiens, seventeen from phlebotomine sandflies (*Phlebotomus* spp.), and one from a hedgehog. All human-derived CHPV genomes retrieved from GenBank were isolated from Indian patients. In contrast, the majority of CHPV genomes from non-human hosts were collected in Senegal (72%, 13/18), followed by Kenya (22%, 4/18), and Nigeria (6%, 1/18) (Figure 1a). The preliminary phylogenetic analysis revealed clear location- and host-dependent clustering patterns, indicating the presence of distinct evolutionary lineages of CHPV (Figure 1b).

Genomes associated with human infections in India between 2003 and 2024 formed a well-supported monophyletic clade, likely reflecting shared geographic and epidemiological origins. This cluster, which includes the 2024 isolates, exhibits genetic variation that may be relevant to viral persistence and transmission dynamics in India. Notably, the 2024 human-derived virus diverges into a separate branch within this clade, suggesting ongoing viral evolution. In contrast, CHPV genomes from non-human hosts, particularly phlebotomine sandflies, demonstrated distinct clustering patterns, likely driven by differences in host-specific selective pressures. Kenyan sandfly isolates (2016–2017) were phylogenetically closer to human-derived CHPV sequences (2003–2024) than to other sandfly-derived viruses, suggesting potential viral exchange between humans and sandflies or shared selective pressures between these two hosts. Conversely, Senegalese sandfly isolates (1978–1997) formed a separate, more divergent cluster, indicating an independent evolutionary trajectory within the sandfly population in this region.

## 3. Discussion

This study enhances our understanding of the genetic diversity of CHPV and underscores the need for further research on its host range and transmission dynamics. By analyzing CHPV genomes from various regions and hosts, the findings provide preliminary insights into the complexity of the CHPV transmission cycle. The data suggest potential interactions between insect vectors and mammalian hosts, but further studies are required to confirm these patterns and assess their epidemiological significance. Sandflies, as highly mobile and abundant vectors, likely play a crucial role in the geographic spread of CHPV, potentially facilitating cross-border transmission. Environmental factors, including climate and vegetation, may further influence sandfly population dynamics, affecting the persistence and transmission of CHPV across affected regions [16]. As a result, outbreaks often coincide with periods of extreme high temperatures (36–49 °C), suggesting a potential impact of climatic conditions on vector behavior and possibly on the clinical phenotype of CHPV [10]. These observations highlight the role of sandflies as a likely vector and potential reservoir for CHPV, emphasizing the need for further research and targeted vector control strategies in endemic regions. The potential adaptation of CHPV to humans is particularly concerning, as it suggests the virus ability to infect and possibly spread within human populations. High contact rates between sandflies and humans in endemic regions, coupled with specific genetic mutations enhancing replication in the human host, likely facilitate this adaptation. Furthermore, the isolation of CHPV from a hedgehog in 1966 highlights its ability to infect a broader range of hosts beyond sandflies and humans, suggesting that hedgehogs may serve as alternative or spillover hosts [12]. Understanding the role of hedgehogs and other mammalian hosts in CHPV ecology could provide valuable insights into how the virus is maintained in nature and the factors that may facilitate potential cross-species transmission.

The ability of CHPV to infect diverse hosts highlights its ecological versatility. The genetic diversity across hosts is likely shaped by differential selective pressures by unique host environments, including the interaction between viral proteins and host immune modulators, as seen in other arboviruses. Studies of vectors suggest that host-virus coevolution is driven by selective pressures on both host immunity and viral replication mechanisms, a phenomenon that may also influence CHPV adaptability to mammalian and insect hosts [17,18,19].

These preliminary findings show CHPV potential to persist and evolve within ecologically diverse systems. Phylogenetic analysis confirms the genetic distinction of the Kenyan strain from West African and Asian strains, suggesting that CHPV is widely distributed among sandflies in these regions [14]. This evolutionary pattern raises questions about whether inter-regional transmission events or a shared origin explain these relationships. Notably, the time gap between genome isolations in Senegal (1966–1997) and Kenya (2016–2017) underscores the need for more robust genomic sampling to bridge these temporal gaps and better understand CHPV evolutionary trajectory. All human-derived CHPV genomes analyzed in this study originated from India, reflecting the geographic concentration of available genomic data. The phylogenetic analysis supports the presence of distinct CHPV lineages across different hosts and geographic locations, suggesting long-term circulation in both human and non-human reservoirs. However, the limited availability of human-derived sequences from other endemic regions restricts broader epidemiological interpretations.

Genomic surveillance plays a pivotal role in monitoring CHPV circulation. Generating new CHPV sequences from both human and non-human hosts is critical to building a comprehensive understanding of the virus evolution, transmission dynamics, and ecological adaptations. The adaptability of CHPV, particularly in regions where humans and sandflies coexist, further emphasizes the need for an integrated approach. Expanding surveillance programs to include both human and non-human populations under a unified One Health framework is essential. By bridging ecological, virological, and epidemiological gaps, genomic surveillance can provide the foundation for proactive measures to mitigate CHPV impact and prevent future outbreaks.

## 4. Materials and Methods

### 4.1. Data Collection and Sequence Selection

A total of 23 complete CHPV genomes, each exceeding 11,000 nucleotides in length and accompanied by metadata on geographic location and date of isolation, were retrieved from the GenBank database (NCBI). The selected genomes spanned diverse regions, hosts, and time periods to enable a robust analysis of CHPV evolutionary dynamics. Table 1 reports the characteristics of the 23 genomes.

### 4.2. Sequence Alignment and Editing

Sequences were aligned using the Multiple Alignment using Fast Fourier Transform (MAFFT) software version 7 [20], which ensures high accuracy for RNA virus genomes. The alignment was manually inspected and edited in AliView version 1.28 [21] to correct any inconsistencies and remove poorly aligned regions, ensuring high-quality input for downstream analyses.

### 4.3. Preliminary Phylogenetic Analysis

A preliminary phylogenetic tree was constructed using IQ-TREE 2 [22], employing the General Time Reversible model with gamma-distributed rate variation among sites (GTR + G4). The GTR + G4 model was selected based on the ModelFinder tool integrated within IQ-TREE 2, which identifies the best-fitting substitution model for the dataset.

### 4.4. Temporal Signal Assessment

To assess the suitability of the dataset for time-scaled phylogenetic analysis, the molecular clock signal strength was evaluated using root-to-tip regression in TempEst v1.5.3 [23]. Sequences identified as potential outliers based on significant deviations from the clock assumption were removed to improve the reliability of subsequent temporal analyses.

### 4.5. Time-Scaled Phylogenetic Analysis

Time-scaled phylogenetic trees were inferred using Bayesian Evolutionary Analysis Sampling Trees (BEAST) v1.10.4 [24]. A relaxed lognormal molecular clock model was applied to account for rate variation among branches, combined with a Bayesian skyline coalescent prior to model population size changes over time. The GTR substitution model was used based on ModelFinder results. The MCMC chain was run for 10 million iterations, sampling every 1000 iterations. An empirical distribution of 1000 trees was generated and assessed for effective sample sizes (ESS) to confirm sufficient sampling from the posterior distribution. Maximum clade credibility (MCC) trees were summarized using TreeAnnotator v1.10.4 [24] after removing a 10% burn-in to account for parameter equilibration. MCC trees were visualized and annotated using the ggtree package in R Studio version 4.4.2 [25], providing a clear representation of CHPV evolutionary relationships and temporal dynamics.

## 5. Conclusions

In conclusion, this study provides preliminary insights into the evolutionary dynamics of CHPV, revealing distinct lineages and transmission dynamics among different hosts and geographic regions. Our findings highlight the importance of expanded genomic surveillance and comprehensive phylogenetic analysis to understand the adaptation and spread of CHPV. By integrating data from multiple hosts and locales, we can enhance our understanding of CHPV epidemiological complexities, potentially guiding targeted public health interventions. Future research should focus on filling the gaps in genomic data and exploring the role of environmental and host-specific factors in shaping the virus evolution and host interactions.

## Figures and Tables

**Figure 1 ijms-26-01021-f001:**
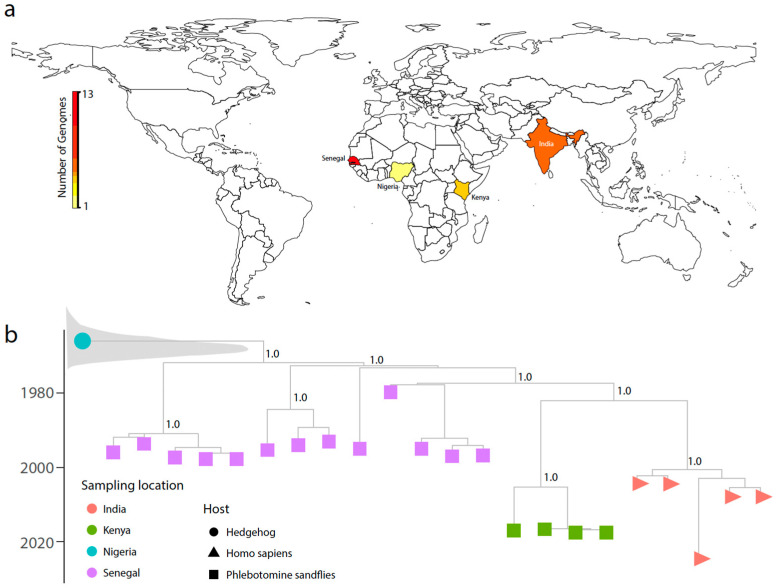
Geographic distribution and phylogenetic analysis of *Chandipura virus* (CHPV) genomes. (**a**) Map showing the number of CHPV genomes available from different countries, highlighted using a color gradient from yellow (1 genome) to red (13 genomes). Countries with CHPV genome sequences include India (13), Senegal (1), Nigeria (3), and Kenya (2); (**b**) Time-scaled phylogenetic tree of CHPV genomes inferred using BEAST v1.10.4. Sampling locations are indicated by different colored circles: red for India, green for Kenya, teal for Nigeria, and purple for Senegal. Host types are represented by different shapes: triangles for humans, squares for phlebotomine sandflies, and circles for hedgehogs. The tree is based on a relaxed lognormal clock model with 10 million MCMC iterations, showing posterior probabilities for major clades, all of which have strong support (posterior probability = 1.0). The phylogeny illustrates the temporal and geographic diversification of CHPV across different hosts and regions.

**Table 1 ijms-26-01021-t001:** Characteristics of *Chandipura virus* (CHPV) genome sequences collected from GenBank (NCBI), including accession number, location, year and host of collection.

Accession Number	Location	Year	Host
HM627186.1	Nigeria	1966	Hedgehog
PQ185534.1	India	2024	*Homo sapiens*
GU190711.1	India	2007	*Homo sapiens*
GU212858.1	India	2003	*Homo sapiens*
GU212856.1	India	2004	*Homo sapiens*
GU212857.1	India	2007	*Homo sapiens*
MT019619.1	Senegal	1997	*Phlebotomine sandflies*
MT019618.1	Senegal	1995	*Phlebotomine sandflies*
MT019617.1	Senegal	1995	*Phlebotomine sandflies*
MT019616.1	Senegal	1997	*Phlebotomine sandflies*
MT019615.1	Senegal	1995	*Phlebotomine sandflies*
MT019614.1	Senegal	1997	*Phlebotomine sandflies*
MT019613.1	Senegal	1997	*Phlebotomine sandflies*
MT019612.1	Senegal	1997	*Phlebotomine sandflies*
MT019611.1	Senegal	1995	*Phlebotomine sandflies*
MT019610.1	Senegal	1994	*Phlebotomine sandflies*
MT019609.1	Senegal	1992	*Phlebotomine sandflies*
MT019608.1	Senegal	1995	*Phlebotomine sandflies*
ON158119.1	Kenya	2017	*Phlebotomine sandflies*
ON158118.1	Kenya	2016	*Phlebotomine sandflies*
ON158117.1	Kenya	2017	*Phlebotomine sandflies*
ON158116.1	Kenya	2016	*Phlebotomine sandflies*
HM627187.1	Senegal	1978	*Phlebotomine sandflies*

## Data Availability

The sequencing data are openly available on GenBank (NCBI) [26]. Other data are available upon reasonable request.

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
