# Peer review of "Phylogenetic Analysis of Chandipura virus: Insights from a Preliminary Genomic Study"

_ijms, 2025, doi:10.3390/ijms26031021_

Round 1
Reviewer 1 Report
Comments and Suggestions for Authors
Overall comments:
This paper provides valuable insights into the evolutionary dynamics of Chandipura virus (CHPV), particularly its host-specific evolution and potential for adaptation across regions and species. Genomic analysis and phylogenetic analysis provided clues of how geographic and host-specific factors shape the virus’s evolution. The study makes significant contributions to understanding the role of sandflies as vectors and suggests the potential for viral exchange between humans and insect populations. However, the study could benefit from detail genomic comparison analysis, such as further functional analysis of genetic mutations, a more comprehensive sampling across regions and temporal gaps, and expanded diagnostic tool discussions. Here are some suggestions to enhance the study's impact.
In the introduction section: This section provided detailed information about CHPV, however, more information about those collected CHPV genomes are required and the authors should point out the importance of this research.
In the method section: This section needs to be re-written to provide a clearer understanding for readers, especially how the time-scaled phylogenetic analysis was conducted and the rationale for using specific models. The lack of detailed descriptions of genomic quality control measures and data preprocessing may limit the reproducibility of the results for others trying to replicate the analysis.
In the result section: Although the paper provides strong phylogenetic insights, it lacks detailed exploration of the functional impact of the observed mutations. Understanding how the virus's genetic diversity affects pathogenicity, transmissibility, or immune evasion would add depth to the findings. The phylogenetic analysis with the limited number of genomes analyzed (23) may not provide a comprehensive picture of CHPV’s global diversity. Expanding the dataset could improve the overall quality of the analysis.
In the discussion section: the information about host-virus interactions for CHPV’s is limited in the previous part and more evidence required to emphasize the interaction. More detailed information about those viral genomes should be provided for genetic diversity analysis. The adaptability of CHPV also required more information such as the reported 62 individuals.
Besides, some sentence needs to be re-written to make it clear for readers, such as the following:
Line121: “11.000 nucleotides“ should be “11,000 nucleotides”
Line136: “one hedgehog” should be “one from hedgehog”.
Lines 211-213: the statement for cross-species transmission is overstated.
Lines 218-230: more information required to support the Genomic surveillance.
Lines 231-235: Those sentences are meaningless in the discussion section.
Author Response
Overall comment: This paper provides valuable insights into the evolutionary dynamics of Chandipura virus (CHPV), particularly its host-specific evolution and potential for adaptation across regions and species. Genomic analysis and phylogenetic analysis provided clues of how geographic and host-specific factors shape the virus’s evolution. The study makes significant contributions to understanding the role of sandflies as vectors and suggests the potential for viral exchange between humans and insect populations. However, the study could benefit from detail genomic comparison analysis, such as further functional analysis of genetic mutations, a more comprehensive sampling across regions and temporal gaps, and expanded diagnostic tool discussions. Here are some suggestions to enhance the study's impact.
Reply: Thank you for the thoughtful feedback. We appreciate the acknowledgment of our study's contributions and the constructive suggestions. We will incorporate your recommendations where feasible, particularly expanding discussions on genomic comparisons, addressing temporal gaps, and diagnostic tools to enhance the study's scope and impact.
Comment 1: In the introduction section: This section provided detailed information about CHPV, however, more information about those collected CHPV genomes are required and the authors should point out the importance of this research.
Reply: Thank you for the suggestion. We will revise the introduction to include more detailed information about the collected CHPV genomes, such as their geographic and temporal distribution, and emphasize the importance of this research in understanding CHPV’s evolutionary dynamics and its potential public health implications.
Comment 2: In the method section: This section needs to be re-written to provide a clearer understanding for readers, especially how the time-scaled phylogenetic analysis was conducted and the rationale for using specific models. The lack of detailed descriptions of genomic quality control measures and data preprocessing may limit the reproducibility of the results for others trying to replicate the analysis.
Reply: Thank you for this important feedback. We will revise the methods section to provide a clearer and more detailed explanation of the time-scaled phylogenetic analysis, including the rationale for selecting specific models. Additionally, we will include comprehensive descriptions of the genomic quality control measures and data preprocessing steps to enhance the transparency and reproducibility of our study.
Comment 3: In the result section: Although the paper provides strong phylogenetic insights, it lacks detailed exploration of the functional impact of the observed mutations. Understanding how the virus's genetic diversity affects pathogenicity, transmissibility, or immune evasion would add depth to the findings. The phylogenetic analysis with the limited number of genomes analyzed (23) may not provide a comprehensive picture of CHPV’s global diversity. Expanding the dataset could improve the overall quality of the analysis.
Reply: Thank you for your feedback. While we agree that exploring the functional impact of observed mutations would provide valuable insights, this was not the primary aim of our study, which focused on phylogenetic and evolutionary dynamics. Additionally, there is currently a significant paucity of whole-genome sequences (WGS) for CHPV, which hampers the ability to fully assess its global genetic diversity. Expanding the dataset is indeed an important future direction, but our analysis provides a foundational step toward understanding CHPV’s evolutionary patterns and highlights the need for broader genomic surveillance efforts.
Comment 4: In the discussion section: the information about host-virus interactions for CHPV’s is limited in the previous part and more evidence required to emphasize the interaction. More detailed information about those viral genomes should be provided for genetic diversity analysis. The adaptability of CHPV also required more information such as the reported 62 individuals.
Reply: Thank you for your feedback. We based our study on the 23 complete genomes of Chandipura virus (CHPV) publicly available. To the best of our knowledge, genetic information regarding the virus that caused the disease in the WHO reported individuals are not publicly available to be analysed. In addition, no further clinical-epidemiological details regarding those individuals have been released. These were the only genomes available from a public database at the time of the study, representing a diverse geographic and host range, which we used to preliminary explore CHPV dynamics. While the limited dataset constrains the depth of the genetic diversity and host-virus interaction analyses, it highlights a critical gap in CHPV genomic surveillance. We added: “The ability of CHPV to infect diverse hosts highlights its ecological versatility. The genetic diversity across hosts is likely shaped by differential selective pressures by unique host environments, including the interaction between viral proteins and host immune modulators, as seen in other arboviruses. Studies of viral vectors suggest that host-virus coevolution is driven by selective pressures on both host immunity and viral replication mechanisms, a phenomenon that may also influence CHPV adaptability to mammalian and insect hosts [15-17]”
Comment 5: Besides, some sentence needs to be re-written to make it clear for readers, such as the following:
Line121: “11.000 nucleotides“ should be “11,000 nucleotides”
Reply: Done.
Line136: “one hedgehog” should be “one from hedgehog”.
Reply: Done.
Lines 211-213: the statement for cross-species transmission is overstated.
Reply: Done.
Lines 218-230: more information required to support the Genomic surveillance.
Reply: Done.
Lines 231-235: Those sentences are meaningless in the discussion section.
Reply: Done.
Reviewer 2 Report
Comments and Suggestions for Authors
Recently, I reviewed a study entitled “ Phylogenetic Analysis of Chandipura Virus: Insights from a Preliminary Genomic Study,” presented by Marta Giovanetti, Valeria Micheli, Alessandro Mancon, Davide Mileto and Alberto Rizzo. I could recommend to shorted the manuscript to the “letter” in case that it was new material presented by the authors, but in this case, I oppositely recommend to write it as a review article for the reason it relies solely on publicly available data rather than the authors’ original research.
Titles of sections are not corresponding to the demand of the journal and have to be changed.
Abstract. Abstract should include the next structure: introduction, materials and methods, results, discussion/ conclusion. Authors begin from conclusions and the aim of the study. I suggest to re-arrange this section.
Lines 12-13. “This preliminary study investigates the possible evolutionary dynamics of Chandipura virus (CHPV)”. This phrase should be moved to the end of the abstract.
Lines 15-17. “total of 23 CHPV genomes, including isolates from humans, sandflies, and a hedgehog, were analyzed using phylogenetic analyses.” Since this data was received from publicly available database, it should be signed in the abstract.
Lines 23-26. “indicating possible viral exchange or shared selective pressures between humans and sandflies. The study confirms the virus ability to infect both insect vectors and mammalian hosts, including humans and hedgehogs, and emphasizes the importance of surveillance and genomic studies, in both humans and non humans, to track its evolution.” I strongly disagree with the conclusion. Authors compare 1. Indian viruses collected from the clinical human cases only from different years. 2. Global available data on this virus from different dates and geographic regions. Current available data not allows to make such conclusions.
Keywords.
Chandipura to add ”virus”
Introduction.
The introduction section is not suit for the aims of the study, which is genetic/phylogenetic study of the CHPV. There are lot of redundant information on diagnostic technics. The type of the article is “communication”- is a short and strait type. It should be significantly shortened.
The aim of the study is not defined.
Line 31. Genus to which the virus belongs, should be added.
Line 46. “observed during outbreaks” it is redundant phrase. To delete.
Lines 48-53. This information or more suitable for the discussion section. I suggest to shorten it.
Lines 70-71’ “2. Diagnostic Challenges of Chandipura Virus and Genomic Surveillance”. Is there a part of introduction section? What is the importance of whole this data for the aim of the article? A suggest to delete whole the section and to add couple of sentences to the introduction section.
Table 2. It is not result of the study. It is data on sequences from the GeneBank. Maximum were this information may be presented in the supplementary, but not in the “results” section.
Lines 157-159. “Genomes associated with human infections collected in India between 2003 and 2024 formed a well-supported monophyletic clade, suggesting that these viruses may represent a unique evolutionary lineage.” I strongly disagree with such conclusions. It only shows geographic origin of the viruses. The statement has to be corrected.
Lines 170-172. “These findings support the hypothesis that CHPV evolution is shaped by both geographic and host-specific factors, with potential implications for its transmission dynamics and host adaptation across different regions and species.” I strongly disagree with such conclusions. There is no data on sequences from the human cases from Africa, as well as for other host from India. There is no confirmation of such hypotheses. To delete.
Discussion.
Lines 175-176. “The current study provides valuable insights into the evolutionary dynamics and host adaptability of CHPV”. with this conclusion since absence of confirmations. To rewrite.
Lines 177-179. By analyzing CHPV genomes from various regions and hosts, the findings underscore the complexity of the CHPV transmission cycle and its remarkable ability to adapt to diverse environments and species”. I strongly disagree with this conclusion since absence of confirmations. To rewrite.
Lines 180-181. “as the data demonstrate the virus's adaptability to both insect vectors and mammalian hosts”. the same. To delete or rewrite.
Lines 200-205. “Outbreaks often coincide with periods of extreme high temperatures (36–49 °C), highlighting the potential impact of climatic conditions not only on vector behavior but also on the clinical phenotype of CHPV [7]. These findings confirm sandflies as the primary vector and significant reservoir for CHPV, underscoring the need for targeted vector control strategies in endemic regions.” the same. To delete or rewrite.
Reference.
Reference section does not meet the requirements of the journal.

Author Response
Overall comment: Recently, I reviewed a study entitled “Phylogenetic Analysis of Chandipura Virus: Insights from a Preliminary Genomic Study,” presented by Marta Giovanetti, Valeria Micheli, Alessandro Mancon, Davide Mileto and Alberto Rizzo. I could recommend to shorted the manuscript to the “letter” in case that it was new material presented by the authors, but in this case, I oppositely recommend to write it as a review article for the reason it relies solely on publicly available data rather than the authors’ original research.
Reply: Thank you for your feedback. We appreciate your suggestion to consider a review format; however, our study focuses on presenting novel phylogenetic insights and methodological approaches using publicly available data, emphasizing the significant gaps in genomic surveillance for CHPV. We believe these findings merit their current format to highlight the urgent need for further research and data generation.
Comment 1: Titles of sections are not corresponding to the demand of the journal and have to be changed.
Reply: Done.
Comment 2: Abstract. Abstract should include the next structure: introduction, materials and methods, results, discussion/ conclusion. Authors begin from conclusions and the aim of the study. I suggest to re-arrange this section.
Reply: Done.
Comment 3: Lines 12-13. “This preliminary study investigates the possible evolutionary dynamics of Chandipura virus (CHPV)”. This phrase should be moved to the end of the abstract.
Reply: Done.
Comment 4: Lines 15-17. “total of 23 CHPV genomes, including isolates from humans, sandflies, and a hedgehog, were analyzed using phylogenetic analyses.” Since this data was received from publicly available database, it should be signed in the abstract.
Reply: Done.
Comment 5: Lines 23-26. “indicating possible viral exchange or shared selective pressures between humans and sandflies. The study confirms the virus ability to infect both insect vectors and mammalian hosts, including humans and hedgehogs, and emphasizes the importance of surveillance and genomic studies, in both humans and non humans, to track its evolution.” I strongly disagree with the conclusion. Authors compare 1. Indian viruses collected from the clinical human cases only from different years. 2. Global available data on this virus from different dates and geographic regions. Current available data not allows to make such conclusions.
Reply: Thank you for your comment. We acknowledge the dataset limitations and will revise the text to clarify that these are preliminary observations, not definitive conclusions. We will also emphasize the need for expanded genomic data to further investigate these hypotheses.
Comment 6: Keywords: Chandipura to add ”virus”
Reply: Done.
Comment 7: Introduction. The introduction section is not suit for the aims of the study, which is genetic/phylogenetic study of the CHPV. There are lot of redundant information on diagnostic technics. The type of the article is “communication”- is a short and strait type. It should be significantly shortened.
The aim of the study is not defined.
Reply: Thank you for your feedback. We will revise the introduction to align more closely with the study's genetic and phylogenetic focus, remove redundant information on diagnostic techniques, and clearly define the study's aim. The section will be streamlined to suit the concise format of a communication article.
Comment 8: Line 31. Genus to which the virus belongs, should be added.
Reply: Done.
Comment 9: Line 46. “observed during outbreaks” it is redundant phrase. To delete.
Reply: We appreciate the reviewer feedback; we have now revised the entire section.
Comment 10: Lines 48-53. This information or more suitable for the discussion section. I suggest to shorten it.
Reply: Done.
Comment 11: Lines 70-71’ “2. Diagnostic Challenges of Chandipura Virus and Genomic Surveillance”. Is there a part of introduction section? What is the importance of whole this data for the aim of the article? A suggest to delete whole the section and to add couple of sentences to the introduction section.
Reply: Thank you for your comment. Further details regarding CHPV, which include diagnostics, were included at the editor's request during initial submission. However, we will revise it to ensure its relevance to the article's aim and add linking sentences to the introduction to better connect the diagnostic challenges, genomic surveillance, and the study's objectives.
Comment 12: Table 2. It is not result of the study. It is data on sequences from the GeneBank. Maximum were this information may be presented in the supplementary, but not in the “results” section.
Reply: Done. We moved table 2 to the Materials and Methods section.
Comment 13: Lines 157-159. “Genomes associated with human infections collected in India between 2003 and 2024 formed a well-supported monophyletic clade, suggesting that these viruses may represent a unique evolutionary lineage.” I strongly disagree with such conclusions. It only shows geographic origin of the viruses. The statement has to be corrected.
Reply: Thank you for the feedback. We agree that the phrasing could lead to misinterpretation. We have revised the statement to clarify that the observed clustering primarily reflects the geographic origin of the viruses rather than suggesting a distinct evolutionary lineage.
Comment 14: Lines 170-172. “These findings support the hypothesis that CHPV evolution is shaped by both geographic and host-specific factors, with potential implications for its transmission dynamics and host adaptation across different regions and species.” I strongly disagree with such conclusions. There is no data on sequences from the human cases from Africa, as well as for other host from India. There is no confirmation of such hypotheses. To delete.
Reply: Thank you for pointing this out. We agree that the available data does not provide sufficient evidence to support this hypothesis conclusively. As suggested, we have removed the statement to ensure the conclusions remain aligned with the presented data.
Discussion.
Comment 15: Lines 175-176. “The current study provides valuable insights into the evolutionary dynamics and host adaptability of CHPV”. with this conclusion since absence of confirmations. To rewrite.
Reply: Thank you for the feedback. We agree that the conclusion needs to be more cautious given the limitations of the data. We have revised the statement to: "The current study contributes to understanding the genetic diversity and potential evolutionary trends of CHPV, while highlighting the need for further research to confirm host adaptability and transmission dynamics."
Comment 16: Lines 177-179. By analyzing CHPV genomes from various regions and hosts, the findings underscore the complexity of the CHPV transmission cycle and its remarkable ability to adapt to diverse environments and species”. I strongly disagree with this conclusion since absence of confirmations. To rewrite.
Reply: Thank you for your comment. We recognize that the conclusion may overstate the implications of the findings. We have now revised the statement.
Comment 17: Lines 180-181. “as the data demonstrate the virus's adaptability to both insect vectors and mammalian hosts”. the same. To delete or rewrite.
Reply: Thank you for your observation. We agree that this statement lacks sufficient evidence to make such a definitive conclusion. We have revised the sentence to: "The data suggest the virus's potential for adaptability to both insect vectors and mammalian hosts, but further studies are needed to confirm this hypothesis."
Comment 18: Lines 200-205. “Outbreaks often coincide with periods of extreme high temperatures (36–49 °C), highlighting the potential impact of climatic conditions not only on vector behavior but also on the clinical phenotype of CHPV [7]. These findings confirm sandflies as the primary vector and significant reservoir for CHPV, underscoring the need for targeted vector control strategies in endemic regions.” the same. To delete or rewrite.
Reply: Thank you for your comment. We agree that the statement requires revision to avoid overinterpretation. We have rephrased it as follows: "Outbreaks often coincide with periods of extreme high temperatures (36–49 °C), suggesting a potential impact of climatic conditions on vector behavior and possibly on the clinical phenotype of CHPV [7]. These observations highlight the role of sandflies as a likely vector and potential reservoir for CHPV, emphasizing the need for further research and targeted vector control strategies in endemic regions."
Comment 19: Reference. Reference section does not meet the requirements of the journal.
Reply: We revised references format according to IJMS Instructions for Authors.
Round 2
Reviewer 2 Report
Comments and Suggestions for Authors
Recently, I reviewed a study entitled “ Phylogenetic Analysis of Chandipura Virus: Insights from a Preliminary Genomic Study,” presented by Marta Giovanetti, Valeria Micheli, Alessandro Mancon, Davide Mileto and Alberto Rizzo. The authors neglected many my recommendation from my previous review and didn’t presented explanation for it. I still believe that this type of article is and not communication for the reason the basis of the study from publicly available sources and suitable to be published in format of “mini review” since it does not present a complete information neither on virus, not on pathology and other aspects of the virus/disease/affects. The manuscript includes presentation of diagnostic methods (introduction, Table 1), when the title of the manuscript staid the same: “ Phylogenetic Analysis of Chandipura Virus: Insights from a Preliminary Genomic Study “. In case of communication format, this table and almost all information on diagnostic methods should be deleted.
Order of sections is not meet the requirements of the journal.
Nomenclature of species, genera, families should be verified which fond the should be presented: regular or Italic.
Abstract.
The word “Introduction”, “Materials and Methods”, “Results” and “Discussion” are redundant. To delete.
Introduction.
Whole introduction section has to be re-organized for not convenient reading. Point by point, theme by theme.
First paragraph. Lines 30- 36. Authors mixed information on classification, a little bit about history of virus discovery, and clinical signs and history of outbreaks.
2nd paragraph. Lines 37-53. Mixed information on clinical manifestations of CHPV and vectors. After than the author present discussion on diagnosis and the vector, lines 44-47: “The rapid progression of CHPV infections underscores the urgent need for early detection, vaccine development, and therapeutic solutions. Sandflies (Phlebotominae) are recognized as the main vector of CHPV, with evidence of transovarial (vertical) transmission maintaining the virus within sandfly populations.” Further, the authors continue to present information on other probable vectors for the disease.
3d paragraph. Lines 54-61. In this part the authors describe outbreak 2023. From line 61 to lines 64 the theme was switched to the aim study, where “In this study, genomic data from 23 publicly available CHPV sequences, spanning geographic regions (India, Senegal, Kenya, and Nigeria), multiple hosts (humans, sandflies, and hedgehogs), and decades (1965–2024), were analysed to investigate the virus's evolutionary dynamics. “, which should finalize the introduction section and I suggest to move it to the final paragraph of the “introduction” section.
Lines 64-71 is related to discussion section. To transfer it or delete.
Paragraph 4th. Lines 72-100. After beginning with diagnostic challenges. Lines 74-76. The authors switch the attention of the reader next time to clinical signs, which complicate reading. From line 81- come back to diagnosis.
Table 1. Authors neglected my recommendations regarding the Table 1 at all.
1. Table cannot be situated in the introduction section. It the role of whatever article writing.
2. Title. Title in this journal is the title only. All details should be moved to the footling of the table.
3. it is not corresponding to the requests of the journal for table presentation, including Fond, coloring and a vertical line.
Lines 105-120. Is more suitable for discussion section, not the “introduction”.
Results.
Line 124. “most nonhuman genomes, 72% (13/18), were collected in Senegal, followed by Kenya, 22% (4/18),” I suggest to re-write the phrase since it seems like spoken on non-human genomes, but not about genomes of CHPV.
Lines 143-144. “This cluster, which includes the 2024 isolates, may highlight mutations that enhance adaptation to the human host, potentially contributing to sustained”. I still strongly disagree with such conclusions since there are no sequences from host except from human. To re-write.
Lines 151-152. “This finding suggests potential viral exchange between humans and sandflies or shared selective pressures between these”. Authors discuss on different geographic zones and different hosts. The conclusions are wrong. The only conclusion which can be done is origin of the Indian and Kenyan viruses, which probable have the same ancestor.
Discussion.
In general, the section is speculative.
Lines 157-158. “The current study contributes to understanding the genetic diversity and potential evolutionary trends of CHPV”. “potential evolutionary trends”- it is speculations. To delete.
Lines 160-161. “the findings provide preliminary insights into the complexity of the CHPV transmission cycle and its potential for adaptation to diverse environments”. It is speculations. To delete.
Lines 171-174. “All human isolates analyzed in this study originated from India, raising two significant possibilities. First, CHPV may have adapted for more efficient human infection in this region. Alternatively, there might be a paucity of CHPV diagnostics and surveillance in other endemic areas, leading to underreporting.” There is lack of data and it never be filled retrospectively. It is speculations. The only conclusions what can be done basing on this phylogenetic study, is identity/similarity of the viruses, time of the virus appearing and splitting into different lineages/clusters. It should be changed.
Lines 175-176. The emergence of the 2024 genome as a distinct genetic variant supports the hypothesis of an ongoing evolutionary process. There is no mention of this “fact” neither in the results and it is not seen in the phylogenetic tree. In case you didn’t provide information, the sentence should be deleted.
Lines 157-197. There are several subjects in this paragraph and I suggest to divide it for several according to the topic o: epidemiology, vector, discussion of phylogeny.
Lines 197-200. The ability of CHPV to infect diverse hosts highlights its ecological versatility. The genetic diversity across hosts is likely shaped by differential selective pressures by unique host environments, including the interaction between viral proteins and host immune modulators, as seen in other arboviruses.” diversity across hosts”. It is speculations only. There is no prove for this statement. To delete.
Lines 205-207. “Despite the limited number of available CHPV sequences, this study underscores the gap between diagnosed cases and the public availability of completed genomes.“ I disagree with the statement. To delete.
Lines 208-226. Mixed data in epidemiology and diagnosis and prophylactics. To separate these topics.
Reference.
It still not the meet the requirements of the journal.
Line 284. To delete.
Author Response
General comment: Recently, I reviewed a study entitled “ Phylogenetic Analysis of Chandipura Virus: Insights from a Preliminary Genomic Study,” presented by Marta Giovanetti, Valeria Micheli, Alessandro Mancon, Davide Mileto and Alberto Rizzo. The authors neglected many my recommendation from my previous review and didn’t presented explanation for it. I still believe that this type of article is and not communication for the reason the basis of the study from publicly available sources and suitable to be published in format of “mini review” since it does not present a complete information neither on virus, not on pathology and other aspects of the virus/disease/affects. The manuscript includes presentation of diagnostic methods (introduction, Table 1), when the title of the manuscript staid the same: “ Phylogenetic Analysis of Chandipura Virus: Insights from a Preliminary Genomic Study “. In case of communication format, this table and almost all information on diagnostic methods should be deleted. Order of sections is not meet the requirements of the journal. Nomenclature of species, genera, families should be verified which fond the should be presented: regular or Italic.
Reply: Thank you for your feedback and for taking the time to review our manuscript again. We appreciate your comments and acknowledge the importance of addressing all concerns to improve the quality of our work. Below, we provide our responses to the key points raised:
- Lack of Explanation for Previous Recommendations: We apologize if our revisions did not sufficiently address all previous recommendations. We have carefully reviewed our prior responses and have now explicitly integrated the suggested modifications where applicable.
- Manuscript Format (Communication vs. Mini Review): we acknowledge the reviewer perspective regarding the format, we shortened the introduction and removed the presentation of diagnostic methods. Now we believe that our study is best suited for a communication article, as it presents novel genomic insights and phylogenetic analysis. The primary objective of this study is to provide preliminary genomic findings rather than a comprehensive review of the virus, its pathology, or associated clinical aspects. Several studies that analysed publicly available data/genomes have not been published as review/mini-review. An example: doi:10.1128/msphere.00624-23. However, we are open to the editor decision should a reformatting be required.
- Presentation of Diagnostic Methods (Table 1): Given the manuscript focus on phylogenetic analysis, we acknowledge the concern regarding the inclusion of diagnostic methods. We have briefly introduced the diagnostic methods in the manuscript and removed most parts regarding the diagnosis, including table 1.
- Order of Sections: We appreciate the comment regarding the structure and have now adjusted the order of sections to fully comply with the journal’s formatting requirements.
- Nomenclature and Formatting (Species, Genera, Families): We have carefully reviewed and corrected the nomenclature, ensuring proper formatting (regular vs. italic) in accordance with established taxonomic conventions. We used italics for viral taxa at the level of family and below, as reported by U.S. CDC: https://wwwnc.cdc.gov/eid/page/scientific-nomenclature. We applied the same indication for other organisms cited in the manuscript.
Abstract
Comment 1: The word “Introduction”, “Materials and Methods”, “Results” and “Discussion” are redundant. To delete.
Reply: Done.
Introduction.
Whole introduction section has to be re-organized for not convenient reading. Point by point, theme by theme.
Comment 1: First paragraph. Lines 30- 36. Authors mixed information on classification, a little bit about history of virus discovery, and clinical signs and history of outbreaks.
Reply: Edited as required. The introduction section now follows the scheme: virus information, disease and clinical manifestations, outbreak, vectors, aim of the study.
Comment 2: 2nd paragraph. Lines 37-53. Mixed information on clinical manifestations of CHPV and vectors. After than the author present discussion on diagnosis and the vector, lines 44-47: “The rapid progression of CHPV infections underscores the urgent need for early detection, vaccine development, and therapeutic solutions. Sandflies (Phlebotominae) are recognized as the main vector of CHPV, with evidence of transovarial (vertical) transmission maintaining the virus within sandfly populations.” Further, the authors continue to present information on other probable vectors for the disease.
Reply: Edited as required. The introduction section now follows the scheme: virus information, disease and clinical manifestations, outbreak, vectors, aim of the study.
Comment 3: 3d paragraph. Lines 54-61. In this part the authors describe outbreak 2023. From line 61 to lines 64 the theme was switched to the aim study, where “In this study, genomic data from 23 publicly available CHPV sequences, spanning geographic regions (India, Senegal, Kenya, and Nigeria), multiple hosts (humans, sandflies, and hedgehogs), and decades (1965–2024), were analysed to investigate the virus's evolutionary dynamics. “, which should finalize the introduction section and I suggest to move it to the final paragraph of the “introduction” section.
Reply: Thank you for your suggestion. We agree that the transition between the outbreak description and the study aim could be improved. To enhance the logical flow of the introduction, we have now moved the sentence outlining the study aim to the final paragraph of the introduction section, as recommended. The introduction section now follows the scheme: virus information, disease and clinical manifestations, outbreak, vectors, aim of the study.
Comment 4: Lines 64-71 is related to discussion section. To transfer it or delete.
Reply: Done.
Comment 5: Paragraph 4th. Lines 72-100. After beginning with diagnostic challenges. Lines 74-76. The authors switch the attention of the reader next time to clinical signs, which complicate reading. From line 81- come back to diagnosis.
Reply: The entire text has been reorganized to make it more cohesive and connected. The introduction section now follows the scheme: virus information, disease and clinical manifestations, outbreak, vectors, aim of the study.
Comment 6: Table 1. Authors neglected my recommendations regarding the Table 1 at all.
- Table cannot be situated in the introduction section. It the role of whatever article writing.
- Title. Title in this journal is the title only. All details should be moved to the footling of the table.
- it is not corresponding to the requests of the journal for table presentation, including Fond, coloring and a vertical line.
Reply: We removed Table 1.
Comment 7: Lines 105-120. Is more suitable for discussion section, not the “introduction”.
Reply: We removed the part: “A key challenge is the development of reliable, affordable, and rapid diagnostic assays that can be deployed in low-resource settings, allowing for early identification and inter-vention during outbreaks. Strengthening diagnostic capacity requires not only technological innovation but also enhanced training for healthcare workers, increased public health awareness, and the integration of CHPV screening into existing arboviral surveillance programs. Addressing these diagnostic and screening challenges is essential to mitigate the public health impact of CHPV, especially in endemic regions where it poses a growing threat.”, leaving the second part of the paragraph that introduces the aim of the study.
Results.
Comment 8: Line 124. “most nonhuman genomes, 72% (13/18), were collected in Senegal, followed by Kenya, 22% (4/18),” I suggest to re-write the phrase since it seems like spoken on non-human genomes, but not about genomes of CHPV.
Reply: Done.
Comment 9: Lines 143-144. “This cluster, which includes the 2024 isolates, may highlight mutations that enhance adaptation to the human host, potentially contributing to sustained”. I still strongly disagree with such conclusions since there are no sequences from host except from human. To re-write.
Reply: Done.
Comment 10: Lines 151-152. “This finding suggests potential viral exchange between humans and sandflies or shared selective pressures between these”. Authors discuss on different geographic zones and different hosts. The conclusions are wrong. The only conclusion which can be done is origin of the Indian and Kenyan viruses, which probable have the same ancestor.
Reply: Done.
Discussion. In general, the section is speculative.
Comment 11: Lines 157-158. “The current study contributes to understanding the genetic diversity and potential evolutionary trends of CHPV”. “potential evolutionary trends”- it is speculations. To delete.
Reply: Done.
Comment 12: Lines 160-161. “the findings provide preliminary insights into the complexity of the CHPV transmission cycle and its potential for adaptation to diverse environments”. It is speculations. To delete.
Reply: Done.
Comment 13: Lines 171-174. “All human isolates analyzed in this study originated from India, raising two significant possibilities. First, CHPV may have adapted for more efficient human infection in this region. Alternatively, there might be a paucity of CHPV diagnostics and surveillance in other endemic areas, leading to underreporting.” There is lack of data and it never be filled retrospectively. It is speculations. The only conclusions what can be done basing on thisphylogenetic study, is identity/similarity of the viruses, time of the virus appearing and splitting into different lineages/clusters. It should be changed.
Reply: Done.
Comment 14: Lines 175-176. The emergence of the 2024 genome as a distinct genetic variant supports the hypothesis of an ongoing evolutionary process. There is no mention of this “fact” neither in the results and it is not seen in the phylogenetic tree. In case you didn’t provide information, the sentence should be deleted.
Reply: Done.
Comment 15: Lines 157-197. There are several subjects in this paragraph and I suggest to divide it for several according to the topic o: epidemiology, vector, discussion of phylogeny.
Reply: Done. We thank the reviewer for the comment. We revised those parts of the discussion. First, we discussed the hosts, then the geographic distribution of CHPV. We improved the smoothness of the transitions. However, those elements are strictly related.
Comment 16: Lines 197-200. The ability of CHPV to infect diverse hosts highlights its ecological versatility. The genetic diversity across hosts is likely shaped by differential selective pressures by unique host environments, including the interaction between viral proteins and host immune modulators, as seen in other arboviruses.” diversity across hosts”. It is speculations only. There is no prove for this statement. To delete.
Reply: We thank the reviewer for the comment. However, we added this part following the request from another reviewer who asked to further explore host-virus interactions.
Comment 17: Lines 205-207. “Despite the limited number of available CHPV sequences, this study underscores the gap between diagnosed cases and the public availability of completed genomes.“ I disagree with the statement. To delete.
Reply: We thank the reviewer for comment. However, we believe this statement is critical for emphasizing the need for enhanced genomic sequencing efforts and public data sharing to support public health objectives. While hundreds of CHPV cases have been diagnosed, only around 20 complete genomes are currently available in public databases. This disparity highlights an important gap that needs to be addressed to better understand virus genetic diversity, evolution, and epidemiological dynamics.
Comment 18: Lines 208-226. Mixed data in epidemiology and diagnosis and prophylactics. To separate these topics.
Reply: To improve the readability, we rephrased it as follows: “Genomic surveillance plays a pivotal role in monitoring CHPV circulation. Generating new CHPV sequences from both human and non-human hosts is critical to building a comprehensive understanding of the virus evolution, transmission dynamics, and ecological adaptations. The adaptability of CHPV, particularly in regions where humans and sandflies coexist, further emphasizes the need for an integrated approach. Expanding surveillance programs to include both human and non-human populations under a unified One Health framework is essential. By bridging ecological, virological, and epidemiological gaps, genomic surveillance can provide the foundation for proactive measures to mitigate CHPV impact and prevent future outbreaks”.
Reference.
It still not the meet the requirements of the journal.
Line 284. To delete.
Reply: Done.
